# The FeetMe^®^ Insoles System: Repeatability, Standard Error of Measure, and Responsiveness

**DOI:** 10.3390/s24186043

**Published:** 2024-09-18

**Authors:** Nathan Martin, Fabien Leboeuf, Didier Pradon

**Affiliations:** 1Pôle Parasport—ISPC Synergies, CHU Raymond Poincaré, APHP, 92380 Garches, France; nathanmartin.coe@outlook.fr; 2Service de Médecine Physique et Réadapatation Locomotrice et Respiratoire, CHU Nantes, Nantes Université, 44093 Nantes, France; fabien.leboeuf@chu-nantes.fr; 3Movement-Interactions-Performance (MIP), EA 4334, CHU Nantes, Nantes Université, 44000 Nantes, France; 4U1179 Endicap, UVSQ, 78000 Versailles, France

**Keywords:** gait, insoles, wearable, minimum detectable change

## Abstract

Background: Three-dimensional motion analysis using optoelectronic cameras and force platforms is typically used to quantify gait disorders. However, these systems have various limitations, particularly when assessing patients in an ecological environment. To address these limitations, several wearable devices have been developed. However, few studies have reported metrological information regarding their repeatability and sensitivity to change. Methods: A healthy adult performed 6 min walking tests with FeetMe^®^ system insoles under different walking conditions overground and on a treadmill. The standard error of measurement (SEM), the minimum detectable differences (MDDs), and the effect size (ES) were calculated for spatio-temporal parameters, and the ground reaction force was calculated from the 16,000 steps recorded. Results: SEM values were below 3.9% for the ground reaction force and below 6.8% for spatio-temporal parameters. ES values were predominantly high, with 72.9% of cases between overground and treadmill conditions with induced asymmetry, and 64.5% of cases between treadmill conditions with and without induced asymmetry exhibiting an ES greater than 1.2. The minimum detectable differences ranged from 4.5% to 10.7% for ground reaction forces and 2.1% to 18.9% for spatio-temporal parameters. Conclusion: Our study demonstrated that the FeetMe^®^ system is a reliable solution. The sensitivity to change showed that these instrumented insoles can effectively reflect patient asymmetry and progress.

## 1. Introduction

Three-dimensional (3D) analysis of movement using optoelectronic cameras and force platforms is commonly used to quantify gait parameters for monitoring gait disorders and their therapeutic management [1]. Although three-dimensional motion analysis in a laboratory setting enables physiologically realistic quantification through the use of biomechanical models [2] and repeatable protocols [3], this assessment only reflects a portion of a patient’s locomotor potential. 

Analysis of walking with wearable sensors offers the possibility of quantifying movement in environments typically encountered by a patient, such as slopes, descents, inclines, holes, etc. These sensors also enable the collection of a greater number of steps, making it possible to quantify spatio-temporal fluctuations in order to understand motor control during walking, or to quantify the effect of therapeutic treatments [4,5,6,7,8]. Del Din et al. proposed an analysis of the relevance, validity, and interest of using an inertial measurement unit (IMU) positioned at the lumbar level to quantify the gait of Parkinson’s disease patients [4]. This comparative work on numerous gait parameters highlights that an IMU placed at this location does not provide the precision expected by clinicians but allows the detection of movement situations. According to these authors, the use of embedded sensors facilitates monitoring of movement quantity, particularly for the studied population. This work underscores the scientific community’s interest in evaluating these new technological solutions for clinical use. This point is also emphasized by other authors [1], who recommend, in addition to clinical gait disorder analysis service management, that the reliability of data collected by these solutions is paramount. 

Over the last decade, a number of wearable devices were designed to quantify the spatio-temporal parameters of walking [9]. These authors have analyzed the use and characteristics of existing solutions such as instrumented socks and insoles for quantifying gait parameters and foot pressure. Although this analysis provides researchers with a comprehensive view of existing solutions, it is also necessary for the scientific community to continue testing the practical validity of these solutions independently of their metrological precision.

Among recent devices, instrumented insoles (FeetMe^®^ Monitor, Paris, France; Loadsol^®^ Novel, Munich, Germany; and Xsensor^®^, Calgary, Canada) appear to be attractive and reliable solutions for quantifying gait cycle parameters and offer an estimation of ground reaction forces [10,11,12,13,14,15,16,17,18,19,20]. To integrate instrumented insoles into the ecosystem of 3D motion analysis laboratories [20], it is crucial to conduct metrological validations of these sensors in accordance with COSMIN recommendations [21]. McGinley et al. conducted a literature review to evaluate the reliability of joint kinematic measurements using biomechanical models with optoelectronic camera systems [20]. The authors noted two points: First, the average joint angle measurement error is 5°, but this error varies with different joints and degrees of freedom quantified. In evaluating industrial or scientific movement analysis solutions, this study highlights the impact of choosing the appropriate biomechanical model for joint kinematics quantification. It also emphasizes the gap between the sensor metrology’s reliability and the calculated parameters after sensor data processing. Specifically, the marker localization inaccuracy is about 1 mm for many optoelectronic camera-based solutions, whereas the biomechanical model’s inaccuracy is due to simplification assumptions. Users must consider both the solution’s precision and the calculated parameters’ validity within their clinical context.

Recently, ref. [10] conducted a study on the validity of the Feetme^®^ system for spatio-temporal gait cycle parameters. The authors quantified differences in spatio-temporal gait parameters in healthy volunteers using the Feetme^®^ system and a reference system (GAITRite^®^). They found a high concordance between the two systems in measuring these parameters. Whilst the accuracy of FeetMe^®^ insoles were studied [10,12], the literature lacks information about repeatability and sensitivity to change, which are both essential for clinical decision-making [22,23,24,25,26] and patient monitoring [27,28,29,30,31,32]. Boekesteijn et al. conducted a literature review to determine the feasibility of using inertial measurement units to quantify gait disorders in individuals with knee osteoarthritis [28]. Although the authors did not find standardized positioning for the inertial units, their analysis clearly showed that inertial sensors could detect gait disorders in individuals with knee osteoarthritis. According to these authors, spatiotemporal parameters sufficiently reflect gait disorders in this population, making them relevant for monitoring mobility. This study highlights a new aspect of using these technological solutions. Indeed, we have indicated that the metrological validity of sensors and the calculated parameters, such as spatiotemporal gait cycle parameters, are crucial, as evidenced by numerous studies [33,34,35,36,37,38,39,40,41]. However, it is equally important for clinical applications to determine if these solutions can detect changes. This concept is also known as minimal detectable difference (MDD). Wells et al. emphasize its importance in clinical study design and have reviewed methods for quantifying it [25]. 

Induced asymmetry is utilized to simulate gait abnormalities commonly observed in clinical populations, such as stroke patients or individuals with musculoskeletal disorders, who often exhibit asymmetric gait patterns [13,42]. By introducing controlled asymmetry in a healthy individual, we can rigorously assess the FeetMe^®^ system’s capacity to detect and measure variations in gait parameters, which is a crucial factor for its application in clinical settings [27]. This approach enables us to evaluate the system’s responsiveness and reliability in identifying clinically relevant changes, thereby ensuring its effectiveness for monitoring and evaluating therapeutic interventions in patients with gait disorders [25,43].

Our study builds on the scientific community’s work on the validity and reliability of embedded technological solutions for quantifying spatiotemporal gait cycle parameters, particularly in detecting sensitivity to change. The aim of our study is to measure the repeatability of FeetMe^®^ system output data, i.e., estimating ground reaction forces and spatio-temporal parameters, as well as sensitivity to change by using induced-asymmetry walking.

## 2. Materials and Methods

### 2.1. Participant

An asymptomatic adult (60 kg, 1.75 m, 25 years) completed all 24 6 min walking tests, taking 8363 left-foot steps and 8359 right-foot steps. A single model of shoe was used during the different conditions (Kiprun KD500). At the start of each test, the insoles were placed in the subject’s shoes and calibrated according to the FeetMe^®^ recommended calibration procedure [10]. For each walking condition, data were recorded after a 5 min accommodation period on a treadmill. This evaluation was carried out within the framework of a database of healthy subjects for a research protocol on the effect of repeated botulinum toxin injections [Committee for the protection of persons île-de-France 2015-A01671-48, NCT02699775].

### 2.2. Material: Insole System

FeetMe^®^ system insoles consist of an IMU and 18 capacitive cells. The IMU samples at a frequency of 140 Hz and comprises an accelerometer (±8 g) and a gyroscope (±1000 dps). The capacitive cells have a surface area of 15 mm², an 8-bit digital signal, and an acquisition frequency of 110 Hz. The internal algorithm of the insoles uses the pressure values of the capacitive cells to calculate the spatio-temporal parameters of the gait cycle (i.e., duration of the stance phase, duration of the single stance phase, duration of the double stance phase, and duration of the oscillating phase). The detection of initial contact is calculated from the sum of the signal from the sensors in each cell over time (S) and the derivative of the sum of the sensor signals (dS/dt) [10]. A Savitzky–Golay filter is then applied [44]. Initial contact is detected when the derivative of the sum of the sensor signals is greater than 0.2 at time (t) and the sum of the signal from the sensors in each cell is greater than 50 at time (t + 100 ms). Toe-off is detected if the sum of the signals in each cell is less than 30 at time (t) and the sum of the signal from the sensors in each cell is less than 30 at time (t + 100 ms) [10]. All the pressure sensors in the capacitive cells provide the distribution of the ground reaction force along the vertical axis. All the signal processing described above is performed by the internal algorithms of the FeetMe^®^ system insoles as described by Jacobs et al. [10].

### 2.3. Procedure and Data Analysis

We used a dual-belt force-instrumented treadmill (M-Gait^®^, Motek Medical B.V., Amsterdam, The Netherlands) to induce asymmetry by modifying the speed of its belts [45,46,47]. We asked the subject to perform 6 min walk tests (TDM6) in different conditions as described in Table 1 (Table 1). For each walking condition, three TDM6 were performed in order to limit intra-individual variability [20]. These three tests were carried out during the same session, with 5 min rest periods between each test. All walking tests were performed over two consecutive days, ensuring adequate rest periods for the participant to mitigate the impact of fatigue. The order of the different walking conditions was randomized.

All the parameter recordings provided by the FeetMe^®^ system are accessible from two comma-separated values files. We specifically selected the following from among all the available parameters: -Mean force during single stance phase: the ground reaction force in the vertical axis, representing the force exerted by the ground on a body in contact with it; -Stance duration: the duration of the support phase, representing the time during which the foot is in contact with the ground supporting the body’s weight; -Single stance duration: the duration of the single support phase, representing the time when only one foot is in contact with the ground while the other foot is in the air; -Double stance duration: the duration of the double support phase, representing the time when both feet are simultaneously in contact with the ground; -Swing duration: the duration of the oscillating phase, representing the time when the foot is swinging through the air between ground contacts.

All the raw data from the FeetMe^®^ insoles were collected and processed using Python 3.9.7 to create a database stored in a CSV file. A custom Python 3.9.7 module, containing various classes and functions for calculating the parameters listed in Table 2, was developed and is available on GitHub (https://github.com/NathanMartinCOE/semelle_connecte (16 September 2024)). All statistical analyses were performed using R software 4.2.1.

To assess the repeatability of the FeetMe^®^ system outputs, the Intraclass Correlation Coefficient (ICC) was calculated using a two-way mixed-effects model, specifically ICC(3,k). The ICC(3,k) assesses the reliability of the average of multiple measurements. The formulas for these calculations are based on the variance components derived from an ANOVA table, following the guidelines outlined by Shrout and Fleiss [48,49].

The standard error of measurement (*SEM*) was calculated using the following formula [50]:
$$SEM=\sqrt{\frac{\sum_{i=1}^{i=n}\sigma_{i}^{2}}{n}}$$

where *SEM* = standard error of measurement; *n* = the number of tests per condition; and 
$\sigma_{i}^{2}$
 = the variance measured for test *i*.

For the parameter of mean force during the single support phase, the SEM was calculated at each instant of the gait cycle, and then averaged. The SEMs were calculated separately for each leg (right and left), and the lowest and highest average values are reported in the results. The SEMs calculated for the test–retest were reported both in the actual unit and expressed as percentages of the mean values (SEM%). A smaller score indicates a more reliable measurement [51].

To assess the degree of sensitivity to change, the minimum detectable difference (MDD) was calculated at 95% using the following formula [25]:
$${MDD}_{95\%}=1.96\times SEM\times \sqrt{2}$$

where 
${MDD}_{95\%}$
 = minimum detectable difference at 95%; and *SEM* = standard error of measurement. 

The MDD is defined as the minimum change that can occur during the measurement that is not due to random variation. Since it represents the degree of sensitivity to change, the MDD is needed to assess whether or not a real change occurs during the execution of two sessions [52]. The authors report that a 90% confidence interval is commonly used, but some studies employ a 95% confidence interval. We chose to use a stricter 95% confidence interval.

The MDDs were calculated for each parameter in three conditions: (i) TDM6 overground, (ii) TDM6 on a treadmill without induced asymmetry, and (iii) TDM6 on a treadmill with induced asymmetry. For the condition TDM6 on a treadmill with induced asymmetry, the maximum SEM value for all conditions with induced asymmetry was used. To obtain a single MDD value for each parameter, the largest SEM value between the right and left leg was used. Absolute reliability indexes were examined in terms of raw units (MDD) and percentages to the parameter mean value (MDD%) [53].

To assess the magnitude of sensitivity to change, the effect size (ES) was calculated using the following formula [43,54]:
$$ES=\frac{{\bar{x}}_{asym}-{\bar{x}}_{without\;asym}}{\sigma_{without\;asym}}\;$$

where *ES* = effect size; 
${\bar{x}}_{asym}$
 = the mean during walking with induced asymmetry; 
${\bar{x}}_{without\;asym}$
 = the mean during walking without induced asymmetry; and 
$\sigma_{without\;asym}$
 = the standard deviation measured during a walk without induced asymmetry.

In addition, statistical tests were carried out to assess whether asymmetry in different gait parameters was detected when inducing asymmetry on the treadmill. The effect size was used to measure the magnitude of the differences observed [54]. These authors also mention that the absolute effect size is useful when the studied variables have intrinsic significance. In our study, preserving the value of spatiotemporal parameters is crucial as they hold significance for the clinical community. Since we had only one participant, comparison of the time series did not seem relevant to us as the results would be too influenced by the individual walking pattern. Therefore, we decided not to calculate the effect size or any other comparison on the ground reaction forces curves. This is different from spatio-temporal parameters, which are much less related to an individual’s pattern in healthy subjects. The spatio-temporal parameters are more robust and less affected by individual variability, making them more suitable for meaningful analysis and comparison in this context.

## 3. Results

This section is divided by subheadings. It provides a concise and precise description of the experimental results, their interpretation, and the conclusions that can be drawn from the experiment.

### 3.1. Standard Error of Measurement

The SEM% found for the ground reaction forces did not exceed 3.9% for overground walking, 1.6% for treadmill walking without induced asymmetry, and 3.7% for treadmill walking with induced asymmetry (Table 2). The SEM% found for the spatio-temporal parameters did not exceed 6.8% for overground walking, 1.7% for treadmill walking without induced asymmetry, and 2.6% for treadmill walking with induced asymmetry (Table 2).

### 3.2. Repeatability

High repeatability was observed when computing ICC values. More specifically, the ICC values were very good (ICC > 0.80) in most cases (44); good (0.60 < ICC < 0.80) in three cases; moderate (0.40 < ICC < 0.60) in two cases; and poor (ICC < 0.40) in one case (Table 3).

### 3.3. Magnitude of Sensitivity to Change; Effect Size (ES)

Very large ESs were observed when comparing conditions with and without induced asymmetry. More specifically, the ESs were very large (ES > 1.2) in most cases (72.9%); large (0. 8 < ES < 1.2) in four cases (8.3%); moderate (0.5 < ES < 0.8) in seven cases (14.6%); small (0.2 < ES < 0.5) in two cases (4.2%); and no effect size was very small (ES < 0.2) (Table 2). 

### 3.4. Degree of Sensitivity to Change; Minimum Detectable Difference (MDD)

The MDD% values calculated in our study are low for both ground reaction force (range: [4.5 to 10.7%]) and spatio-temporal parameters (range [2.1 to 18.9%]). More specifically, the smallest MDD% (2.1%) was found for the duration of the double stance phase during a treadmill walking test without induced asymmetry and the largest MDD% (18.9%) was found for the duration of the single stance phase during the overground walking test.

## 4. Discussion

Our study provides essential metrological information for the clinical application of FeetMe^®^ system insoles by examining their repeatability and responsiveness to change. Complementing previous accuracy investigations [10,11,12,13], we demonstrate that FeetMe^®^ system insoles offer a reliable solution. Farid et al. quantified the precision and reliability of the Feetme^®^ system compared to a reference system (GAITRite^®^) [11]. The authors demonstrated that this system is reliable for quantifying spatio-temporal gait cycle parameters in a population of 29 adult post-stroke patients. However, they noted a lower Intraclass Correlation Coefficient (ICC) for quantifying the swing phase duration on the paretic side. The authors explained that this result might be related to testing conditions, suggesting that more cycles would be needed for stronger concordance. Our study, aiming to analyze sensitivity to change, was based on more than 16,000 recorded gait cycles from our trials. Granja et al. also focused on the concordance of walking speed, cadence, and stride during a timed 25-foot walk test (T25WT) in adult patients with multiple sclerosis [12]. They observed a high ICC across three patient groups categorized by their Expanded Disability Status Scale (EDSS) scores. This work highlights the relevance of using this embedded solution to monitor disease progression characterized by gait cycle parameter deterioration. Parati et al. focused specifically on the impact of experimental conditions [13]. This work complements previous studies showing that for the Feetme^®^ system compared to a GAITRite^®^ system when the same patient is subjected to a condition that could alter their gait, minimal detectable changes are low and acceptable. Our study complements previous work by indicating that the Feetme^®^ insole system’s sensitivity to changes in spatio-temporal parameters effectively identifies gait variations in individuals.

The minimum detectable difference (MDD) of the spatio-temporal parameters is lower than the differences deemed significant for quantifying changes in various patient populations, such as children with cerebral palsy, patients with Parkinson’s disease, and stroke patients [55,56,57,58]. Yang et al. showed significant differences between the spatio-temporal parameters of the paretic and non-paretic legs in stroke patients, with a 6.18% difference in stance and swing phases [58]. The MDD for ground reaction forces falls within the physiological range of asymmetry observed in healthy subjects (−5 to 5%) [59]. Furthermore, the MDD for spatio-temporal parameters aligns with those detected using an OPTOGait photoelectric cell system (6.0–16.5%) [60]. Our study also confirmed the insoles’ high magnitude of sensitivity to change, with very large effect sizes observed between conditions with and without induced asymmetry. Notably, the use of a dual-band treadmill yielded an effect size that increased with the speed of induced asymmetry.

The standard error of measurement (SEM) [SEM%] (<0.34 kg·cm^−2^, [0.9–3.9%]) for ground reaction forces and (<48 ms [0.1–6.8%]) for spatio-temporal parameters indicated high reliability. These SEM values are comparable to those found by Lee et al. (SEM < 30 ms [2.8–6%]) using the OPTOGait photocell system [60]. For ground reaction forces, the SEM values were also comparable to those reported by Farius et al., who used force platforms [26]. However, the SEM scores cannot be directly compared with the literature due to the use of different units (kg·cm^−2^ vs. the conventional N·kg). Nevertheless, these scores are consistent with the sensor margin of error provided by the manufacturer (±0.85 kg·cm^−2^ above 5 kg·cm^−2^) [10].

The participants’ response to induced asymmetry revealed two trends. Firstly, there was a reduction in the variability of gait parameters as the level of induced asymmetry increased. Secondly, the spatio-temporal parameters evolved differently depending on whether the leg was driven by the increase in speed. The increase in speed tended to reduce the duration of the stance phase, single stance phase, and swing phase for the leg that was driven by the treadmill, while these phases remained constant for the leg that was not driven by the treadmill. The duration of the double support phase decreased similarly for both legs with increased speed.

Our study has several limitations. Only one subject was included, preventing generalization of the observed functional trends. However, this single inclusion does not impact our metrological findings. The SEM values are robust and can be adjusted based on the number of participants and repetitions [61]. A second limitation concerns the reaction force profile. The force profiles used in our study were calculated from raw capacitive cell data, which depend on the configuration of the capacitive cells and are not representative of typical force profiles measurable by a force platform.

## 5. Conclusions

Our study reveals the reliability and high sensitivity to change of FeetMe^®^ system insoles. These instrumented insoles could be integrated into the array of measurement tools available to motion analysis laboratories, thereby broadening the scope of investigations by enabling the assessment of subjects in ecological situations.

## Figures and Tables

**Table 1 sensors-24-06043-t001:** Summary of the walking conditions for the 6-minute walk tests (TDM6).

N°	Condition	Description	Belt Speed (m/s)	Belt Speed (m/s)	Asymmetry
			Right	Left	
1	TDM6-SYM-GROUND	TDM6 overground	N/A	N/A	Symmetric
2	TDM6-SYM-MGAIT	TDM6 treadmill: equal belt speeds	1.2	1.2	Symmetric
3	TDM6-ASYM-MGAIT-16L	TDM6 treadmill: 16% left asymmetry	1.2	1.4 s	+16% left asymmetry
4	TDM6-ASYM-MGAIT-33L	TDM6 treadmill: 33% left asymmetry	1.2	1.6	+33% left asymmetry
5	TDM6-ASYM-MGAIT-50L	TDM6 treadmill: 50% left asymmetry	1.2	1.8	+50% left asymmetry
6	TDM6-ASYM-MGAIT-16R	TDM6 treadmill: 16% right asymmetry	1.4	1.2	+16% right asymmetry
7	TDM6-ASYM-MGAIT-33R	TDM6 treadmill: 33% right asymmetry	1.6	1.2	+33% right asymmetry
8	TDM6-ASYM-MGAIT-50R	TDM6 treadmill: 50% right asymmetry	1.8	1.2	+50% right asymmetry

**Table 2 sensors-24-06043-t002:** Summary of value, standard error of measure (*SEM*), minimal detectable difference (*MDD95*), and effect size for all gait parameters.

Gait Parameters	Overground Walking (TDM6-SYM-GROUND)			Treadmill without Induced Asymmetry (TDM6-SYM-MGAIT)			Treadmill with Induced Asymmetry (TDM6-ASYM-MGAIT)			Effect Size	
	Values Mean (sd)	SEM Value (%)	MDD95 Value (%)	Values Mean (sd)	SEM Value (%)	MDD95 Value (%)	Values Mean (sd)	SEM Value (%)	MDD95 Value (%)	O-TA Value	T-TA Value
**Mean force during single stance phase (kg·cm^−2^)**	8.13 (0.02) 8.75 (0.01)	0.31 (3.8%) 0.34 (3.9%)	0.86 (10.58%) 0.94 (10.74%)	8.32 (0.05) 8.72 (0.04)	0.09 (1.1%) 0.14 (1.6%)	0.25 (3%) 0.39 (4.47%)	8.04 (0.03) 8.6 (0.03)	0.08 (0.9%) 0.31 (3.7%)	0.22 (2.56%) 0.86 (10.23%)	-	-
**Stance duration (ms)**	730.3 (19.2) 744.8 (18.5)	42.89 (5.9%) 47.82 (6.4%)	118.89 (16.28%) 132.55 (17.8%)	706.9 (11.5) 719.5 (11.7)	3.44 (0.5%) 6.14 (0.9%)	9.54 (1.33%) 17.02 (2.41%)	583.8 (9.8) 714.1 (13.4)	0.91 (0.1%) 12.17 (1.8%)	2.52 (0.36%) 33.73 (5.01%)	−7.85 −1.66	−10.7 −0.46
**Single stance duration (ms)**	413.6 (24.2) 428.6 (18.0)	23.99 (5.8%) 29.22 (6.8%)	66.5 (16.08%) 80.99 (18.9%)	395.3 (16.2) 407.5 (20.0)	3.17 (0.8%) 6.3 (1.6%)	8.79 (2.16%) 17.46 (4.42%)	314.9 (10.9) 420.7 (11)	1.8 (0.4%) 8.84 (2.6%)	4.99 (1.19%) 24.5 (7.16%)	−5.78 −0.28	−4.96 0.71
**Double stance duration (ms)**	314.9 (17.9) 315.3 (13.4)	19.28 (6.1%) 19.76 (6.3%)	53.44 (16.97%) 54.77 (17.37%)	310.6 (16.0) 310.9 (12.5)	2 (0.6%) 2.37 (0.8%)	5.54 (1.78%) 6.57 (2.12%)	268.2 (8.6) 297.9 (11.1)	1.43 (0.5%) 5.12 (1.7%)	3.96 (1.45%) 14.19 (4.78%)	−3.51 −1.01	−3.36 −0.79
**Swing duration (ms)**	414.5 (14.3) 429 (13.9)	24.24 (5.8%) 29.04 (6.8%)	67.19 (16.21%) 80.49 (18.76%)	395.7 (11.7) 408.3 (10.4)	2.45 (0.6%) 6.77 (1.7%)	6.79 (1.66%) 18.77 (4.74%)	315.0 (10.9) 420.7 (11.0)	1.77 (0.4) 8.32 (2.5%)	4.91 (1.17%) 23.06 (7.07%)	−7.47 −0.51	−8 1.19

Values are expressed as mean (standard deviation) and *SEM*; *MDDs* are expressed as value (%); effect size is expressed as value. For *SEM*, *MDD*, and effect size, the values indicated are the smallest (1st cell line) and the largest (2nd cell line). Effect sizes were calculated between the overground (*O*) and treadmill with induced asymmetry (*TA*) conditions and between the treadmill without induced asymmetry (*T*) and treadmill with induced asymmetry (*TA*) conditions.

**Table 3 sensors-24-06043-t003:** Intraclass Correlation Coefficient (ICC) for all spatio-temporal gait parameters.

Condition	Leg	Stance Duration	Single Stance Duration	Double Stance Duration	Swing Duration
TDM6-SYM-GROUND	Left	0.989	0.991	0.943	0.95
	Right	0.963	0.948	0.939	0.99
TDM6-SYM-MGAIT	Left	0.996	0.991	0.972	0.978
	Right	0.988	0.978	0.971	0.991
TDM6-ASYM-MGAIT-16L	Left	0.976	0.981	0.941	0.875
	Right	0.453	0.881	0.956	0.988
TDM6-ASYM-MGAIT-33L	Left	0.992	0.994	0.935	0.95
	Right	0.913	0.951	0.936	0.994
TDM6-ASYM-MGAIT-50L	Left	0.999	0.999	0.999	0.999
	Right	0.999	0.999	0.999	0.999
TDM6-ASYM-MGAIT-16R	Left	0.994	0.969	0.982	0.995
	Right	0.997	0.995	0.984	0.983
TDM6-ASYM-MGAIT-33R	Left	0.943	0.995	0.948	0.994
	Right	0.964	0.985	0.975	0.995
TDM6-ASYM-MGAIT-50R	Left	0.904	0.958	0.917	0.995
	Right	0.994	0.995	0.917	0.958

ICC(3,k) values were computed following guidelines outlined by Shrout and Fleiss [48,49].

## Data Availability

The original contributions presented in the study are included in the article, further inquiries can be directed to the corresponding author.

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
