# Peer review of "The FeetMe® Insoles System: Repeatability, Standard Error of Measure, and Responsiveness"

_sensors, 2024, doi:10.3390/s24186043_

Round 1
Reviewer 1 Report
Comments and Suggestions for Authors
Please see the attachment.

NA
Author Response
Dear reviewer,
We truly appreciate your proofreading and your comments. To make it easier to read, we’re going to surround your comments and suggestions in black and write our response in italics.
Q: "the purpose of the study was to investigate the reliability of gait parameters during walking with and without asymmetric speed between the two limbs, using special inertial sensors."
Unfortunately, this is not the aim of our study. As mentioned in the introduction: "The aim of our study is to measure the repeatability of the FeetMe® system outputs data, i.e., the estimated ground reaction forces and spatio-temporal parameters, as well as the sensitivity to change from induced-asymmetry walking."
Indeed, the reliability was calculated using data from a single individual because we are assessing the reliability of the FeetMe Insoles, not the individual. We do not intend to generalize the results of this study to a population, as this is not the objective. The objective is to study the metrology of the FeetMe Insoles from the perspective of their repeatability, their standard errors of measure, and their responsiveness. Therefore, as you mentioned, with a single individual, we are evaluating the internal consistency of the different parameters of the FeetMe Insoles.
We apologize for not adequately explaining the reason for using asymmetric speeds. In our objective to test the responsiveness of the FeetMe Insoles, we needed to induce a change in the walking parameters of our healthy subject. The insoles are ultimately intended to be used as a measurement tool for patients. Numerous studies have shown that patients exhibit gait asymmetries. Therefore, we used increasing levels of asymmetric speeds to explore the responsiveness of the FeetMe Insoles.
TILTLE
Q1.
Thank you very much for your valuable feedback. We apologize for the confusion caused by the terminology in the title. We understand that the terms "standard error of measurement" (SEM) and "repeatability" might seem interchangeable, but they refer to different concepts in the context of measurement reliability.
The SEM reflects the absolute reliability of a measurement by quantifying the extent to which repeated measures of the same subject under identical conditions vary due to random errors. In contrast, "repeatability" specifically refers to the consistency of a measurement when the same method is applied under the same conditions over multiple trials. While SEM is a measure of the precision of individual scores, repeatability often involves the calculation of metrics like the Intraclass Correlation Coefficient (ICC) to assess the consistency of measurements across trials.
To clarify this distinction in our manuscript, we have revised the text accordingly. We have also included the calculation of the ICC for repeatability in the relevant sections, including the Methods, Results, and Discussion. We hope these changes address the concerns you raised.
Q2.
Responsiveness refers to the ability of a measurement tool or instrument to detect clinically significant changes over time, even if those changes are small. It is an important property for assessing the effectiveness of treatments or interventions. Responsiveness indicates how well the instrument can reflect true changes in the construct being measured, distinguishing it from random variations or measurement error.
We apologize for not using consistent terminology throughout the article. In our study, we evaluated responsiveness using the effect size.
Effect size is a statistical measure that quantifies the magnitude of change or difference in a variable of interest. It provides a standardized way of comparing the degree of change across different studies or conditions, irrespective of the sample size. Commonly used measures of effect size include Cohen's d, which is calculated as the difference between two means divided by the pooled standard deviation.
In the context of our study, effect size was employed to assess how well the FeetMe Insoles could detect changes in gait parameters induced by asymmetric speeds. This approach allows us to quantify the sensitivity of the insoles to meaningful changes in gait patterns, which is crucial for their application in clinical settings.
By utilizing effect size, we aimed to provide a clear and interpretable metric of responsiveness. However, we acknowledge the inconsistency in terminology and will strive to maintain clarity and consistency in this new version of our article.
Beaton, D. E., Bombardier, C., Katz, J. N., & Wright, J. G. (2001). A taxonomy for responsiveness. Journal of Clinical Epidemiology, 54(12), 1204-1217.
Husted, J. A., Cook, R. J., Farewell, V. T., & Gladman, D. D. (2000). Methods for assessing responsiveness: a critical review and recommendations. Journal of Clinical Epidemiology, 53(5), 459-468.
ABSTRACT
Q3. We have taken your comment into account and have added the Minimal Detectable Differences (MDDs) to the abstract:
“The standard error of measurement (SEM), the minimum detectable differences (MDD) and the effect size (ES) were calculated for the spatio-temporal parameters and the ground reaction force from the 16,000 steps recorded.”
INTRODUCTION
Q12. We hope that the revisions we have made to the introduction have significantly improved its clarity and readability.
Q13. We have taken your comment into account and have provide some references:
“Indeed, we have indicated that the metrological validity of sensors and the calculated parameters, such as spatiotemporal gait cycle parameters, is crucial, as evidenced by numerous studies [29–33].”
Q14. We have taken your comment into account and have name the authors at the beginning of the paragraph:
“Wells and al. emphasize its importance in clinical study design and have reviewed methods for quantifying it. [23]”
We have also ensured this in other similar sections:
“McGinley and al. conducted a literature review to evaluate the reliability of joint kinematic measurements using biomechanical models with optoelectronic camera systems [18].”
“Boekesteijn and al. conducted a literature review to determine the feasibility of using inertial measurement units to quantify gait disorders in individuals with knee osteo-arthritis. [26]”
Q15. Thank you for your insightful comment. We apologize for not initially providing the rationale behind the choice of induced-asymmetry walking for evaluating the reliability and sensitivity to change of the FeetMe® system. We have taken your comment into account and have added some rational:
“Induced asymmetry is utilized to simulate gait abnormalities commonly observed in clinical populations, such as stroke patients or individuals with musculoskeletal disorders, who often exhibit asymmetric gait patterns [11,34]. By introducing con-trolled asymmetry in a healthy individual, we can rigorously assess the FeetMe® sys-tem’s capacity to detect and measure variations in gait parameters, a crucial factor for its application in clinical settings [25]. This approach enables us to evaluate the sys-tem's responsiveness and reliability in identifying clinically relevant changes, thereby ensuring its effectiveness for monitoring and evaluating therapeutic interventions in patients with gait disorders [23,35].”
We have also revised our objective to clarify that we are using induced asymmetry as a tool to measure the response to changes:
”The aim of our study is to measure the repeatability of the FeetMe® system outputs datas, i.e. the estimating ground reaction forces and spatio-temporal parameters, as well as the sensitivity to change by using induced-asymmetry walking.”
METHODS
Q16. We have taken your comment into account and have added the shoe model:
“A single model of shoe was used during the different conditions (Kiprun KD500).”
Q17. We kindly addressed this point in section 21, as we believe the description of the walking tests is more appropriately placed in the Procedure and Data Analysis section.
Q18. We understand that this sentence may lead to confusion. We have emphasized the fact that this study is derived from the internal databases needed to carry out the No-Barriers study, in order to be able to compare the results obtained by our patients on several clinical tests with those of healthy subjects. We have replaced the sentence with: “This evaluation is part of the “No-Barriers” clinical trial database. The “No-Barriers” study uses hospital-internal databases to compare results between patient groups. The results we present are a specific focus on the use of equipment commonly used during clinical activity and in the “No-Barriers” study (NCT05294068).”
Q19. We have amended the sentence to provide a clearer explanation on this point:
“The detection of the initial contact is calculated from the sum of the signal from the sensors in each cell over time (S) and the derivative of the sum of sensor signals (dS/dt).”
Q20. Thank you for the feedback. The paragraph has been revised to clarify the process of detecting initial contact and toe-off. The updated text describes the criteria and method used for detection in a more accessible manner. Although graphical representations of the internal sensor data are not available, this revision aims to provide a clearer understanding of the detection mechanisms. It is worth noting that this description closely follows the methodology outlined in the original article, ensuring consistency with the established approach. Further clarification can be provided if needed:
“The detection of initial contact is calculated from the sum of the signal from the sensors in each cell over time (S) and the derivative of the sum of sensor signals (dS/dt). [8] A Savitzky-Golay filter is then applied [36]. Initial contact is detected when the derivative of the sum of the sensor signals is greater than 0.2 at time (t) and the sum of the signal from the sensors in each cell is greater than 50 at time (t + 100ms). Toe-off is detected if the sum of the signals in each cell is less than 30 at time (t) and the sum of the signal from the sensors in each cell is less than 30 at time (t + 100ms). [8]”
Q21. Thank you for your comments (17 and 21). We have taken your observations into account and clarified this point as follows:
“For each walking condition, three TDM6 were performed in order to limit intra-individual variability [18]. These three tests were carried out during the same session, with 5-minute rest periods between each test. All walking tests were performed over two consecutive days, ensuring adequate rest periods for the participant to mitigate the impact of fatigue. The order of the different walking conditions was randomized.”
Q22. We have provided brief descriptions of the parameters to clarify them for the readers:
“We specifically selected:
- the ground reaction force in the vertical axis (force exerted by the ground on a body in contact with it),
- the duration of the support phase (time during which the foot is in contact with the ground supporting the body's weight),
- the duration of the single support phase (time when only one foot is in contact with the ground while the other foot is in the air),
- the duration of the double support phase (time when both feet are simultaneously in contact with the ground),
- the duration of the oscillating phase (time when the foot is swinging through the air between ground contacts)
from among all the available parameters.
Q23. We have taken your comment into account and have modified the paragraph to clarify this point:
“(…) then averaged. The SEMs were calculated separately for each leg (right and left), and the lowest and highest average values are reported in the results. The SEMs (…)”
Q24. Thank you for your feedback. To clarify, "at each instant of the gait cycle" means that the
SEM was calculated continuously at every point in time throughout the entire gait cycle. These individual SEM values were then averaged to provide a single representative SEM for the entire cycle. This approach ensures a detailed and comprehensive assessment of variability across the gait cycle.
Feel free to let us know if you need further adjustments.
Q25. Thank you for your question. To clarify, the phrase "... the maximum SEM value for conditions with induced asymmetry was used" refers specifically to the use of the highest SEM value when calculating the MDD for the condition TDM6 on treadmill with induced asymmetry. This ensures that the MDD reflects the worst-case scenario of variability. Consequently, the maximum SEM is used in the calculation, which directly impacts the MDD, meaning that the maximum MDD is indeed used to provide a conservative estimate of measurement reliability.
RESULTS
Q26. We have revised the sentence, thank you for your comment.
Q27. We have revised the entire manuscript to use consistent terminology. Thank you for your comment.
Q28. We have revised the entire manuscript to use consistent terminology. Thank you for your comment.
Q29. Thank you for your observation. Even though there is only one condition for the 'overground' and 'without asymmetry' scenarios, the presence of maximum and minimum SEM values is due to the fact that these conditions involve measurements from both the right and left legs. Therefore, the SEM values are reported separately for each leg, and the maximum and minimum values represent the range of variability observed across both legs. This approach ensures that the data reflect any potential differences between the two legs, providing a more comprehensive analysis.
Q30. Thank you for your comment. In the "with induced asymmetry" condition, the smallest and largest SEM and MDD values reported in the first and second cell lines for each gait parameter refer to the range of values calculated across the different walking speed combinations. Specifically, these values represent the minimum and maximum SEM and MDD observed when considering all six different walking speed combinations used during the tests. This approach ensures that the reported SEM and MDD values encompass the variability across various walking speeds, providing a comprehensive assessment of measurement reliability and sensitivity to change under induced asymmetry conditions. We hope this clarifies the reported values.
Q31. We have revised the description of the table. Thank you for your comment.
Q32. We have revised the description of the table. Thank you for your comment.
Q33. We apologize for the oversight. We have replaced all commas with periods for decimal notation and have also revised the formatting of kg.cm-².
Q34. Thank you for your insightful question. We apologize if this point was not clear in the manuscript. To clarify:
- Effect Size measures the magnitude of an observed effect or difference and provides insight into its practical importance relative to the variability in the data. It helps determine the significance of a change in a real-world context.
- Minimal Detectable Difference (MDD), on the other hand, represents the smallest change that can be reliably detected above measurement noise or variability. It focuses on the precision of measurements and determines whether observed changes are statistically significant and not due to random fluctuations.
Both metrics are crucial for understanding and interpreting responses to change. Effect size informs us about the impact and relevance of a change, while MDD assesses the reliability and detectability of that change.
We will make sure to clarify these distinctions in the revised manuscript to better address this important aspect. Thank you for bringing this to our attention.
We have therefore used consistent terminology throughout the manuscript with:
- Magnitude of sensitivity to change, Effect size (ES)
- Degree of sensitivity to change, Minimum Detectable Difference (MDD)
DISCUSSION
Q35. We have taken your comment into account and have name the authors at the beginning:
“Farid and al. quantified the precision and reliability of the Feetme® system compared to a reference system (GAITRite®) [9].”
Q36. We have taken your comment into account and have name the authors at the beginning:
“Granja et al. also focused on the concordance of walking speed, cadence, and stride during a timed 25-foot walk test (T25WT) in adult patients with multiple sclerosis [10].”
Q37. We have taken your comment into account and have name the authors at the beginning:
“Parati and al. focused specifically on the impact of experimental conditions [11].”
Q38. We have taken your comment into account and have name the authors at the beginning:
“Yang and al. showed significant differences between the spatio-temporal parameters of the paretic and non-paretic legs in stroke patients, with a 6.18% difference in the stance and swing phases [48].”
Q39. We have taken your comment into account and revised the formatting of kg.cm-² as well as N.kg.
Q40. Thank you for your valuable feedback. We apologize for the lack of clarity in the original text, which was due to a translation issue. We have corrected it to ensure better understanding. Here is the revised paragraph:
“The participants' response to induced asymmetry revealed two trends. Firstly, there was a reduction in the variability of gait parameters as the level of induced asymmetry in-creased. Secondly, the spatio-temporal parameters evolved differently depending on whether the leg was driven by the increase in speed. The increase in speed tended to re-duce the duration of the stance phase, single stance phase, and swing phase for the leg that was driven by the treadmill, while these phases remained constant for the leg that was not driven by the treadmill. The duration of the double support phase decreased sim-ilarly for both legs with increased speed.”

Reviewer 2 Report
Comments and Suggestions for Authors
The work is of interest, and shows some interesting results. However, serious deficiencies have been found in the content of the work that prevent the reviewer from becoming aware of the scope of the work carried out by the authors. The main question to clarify is about the calculations made by the authors. The reviewer understands that the FeetMe® system insoles provide a series of parameters, and the authors perform a statistical study of these parameters in a set of experiments performed by a volunteer. In that case, the recommendation would be the rejection of the work because the following reasons:
- The authors have not developed hardware that would fit into a sensor research publication. The authors use a commercial device: FeetMe® system insoles.
- The authors have not developed or investigated software, algorithms or computational models for parameter calculation. The authors use the parameters obtained with the FeetMe® system insoles software.
- The statistical study performed is more suitable for a clinical research journal, rather than a technical research journal.
I would recommend authors submit the work to a journal of clinical interest, rather than a technical research journal.
On the contrary, if the authors have developed and implemented the equations, models and algorithms that allow the calculation of the force and duration parameters of the gait phases, the reviewer would recommend a Major Revision in which the procedures and algorithms used for the calculation of the parameters must be defined in much greater detail. Additionally, greater clarity should be provided on the calculations and parameters used.
Comments to the authors:
1) Correct references on page 2, line 43: "or to quantify the effect of therapeutic treatments [4–6[4]"
2) Reference missing on page 2, line 84: "[conducted a literature review to determine the feasibility of using inertial measurement units to quantify"
3) Put the reference [8] in the correct place on page 3, line 125: "the sensors in each cell is less than 30 at time (t + 100ms). [8] All the pressure sensors in"
4) The authors mention the following on page 3, lines 141-142: "All the calculations were implemented using custom Python scripts, available online 141 (https://github.com/NathanMartinCOE/semelle_connecte)."
To facilitate understanding of the code and data, I recommend including an explanatory text document in the root directory, such as Readme.txt or Readme.docx
5) Section "2.2. Material: Insole System" describes the processing carried out on the signals. The authors must clarify whether said processing is carried out by the functions of the FeetMe® system insoles, or by functions developed specifically by the authors. If they have been implemented by the authors, the procedures and algorithms must be defined in greater detail.
6) The authors indicate on page 3, lines 141-142: "All the calculations were implemented using custom Python scripts". The authors should clarify whether the Python scripts correspond to the calculation of the statistics associated with the forces or durations of the phases in the gait, or instead, they correspond to the calculation of forces and durations of the gait phases, as briefly described in section "2.2. Material: Insole System".
7) In relation to the parameters of section "2.3. Procedure and Data analysis":
- the ground reaction force in the vertical axis,
- the duration of the support phase,
- the duration of the single support phase,
- the duration of the double support pase,
- the duration of the oscillating phase from among all the available parameters
The authors should clarify whether these parameters are estimated by FeetMe® system insoles, or by functions developed by the authors. In the latter case, they must precisely specify the algorithms/models used (see previous comments).
8) In section "2.3. Procedure and Data analysis" these parameters are mentioned:
- the ground reaction force in the vertical axis,
- the duration of the support phase,
- the duration of the single support phase,
- the duration of the double support pase,
- the duration of the oscillating phase from among all the available parameters
However, Table 2 indicates these parameters:
- Mean force during single stance phase (kg.cm-²)
- Stance duration (ms)
- Single stance duration (ms)
- Double stance duration (ms)
- Swing duration (ms)
Authors should use common terminology to avoid mistakes (They use different terms for the same parameters). In addition, it is advisable to include a short description of the meaning of each parameter.
9) The number of steps and the duration of each test are mentioned in the work, but the number of tests performed for each type of test is not mentioned. This must be clarified.
10) A more detailed and precise description of all calculations must be made. For example: For the calculation of SEM related to "stance duration" perhaps the authors want to say the following: for each gait cycle the stance duration is calculated, and for the entire 6-minute experiment the mean value and variance of the stance duration are calculated. Finally, it is understood that the formula on line 147 on page 4 is used, making a square mean of the standard deviation of all the tests corresponding to a condition. It's right? The procedure must be clarified and specified in detail for all calculated parameters.
11) In relation to the previous question: Table 2 indicates the mean value and standard deviation of the parameters, SEM and MDD95. The authors should clarify precisely how these parameters are calculated.
12) Since SEM and MDD95 maintain a relationship according to the equation on line 158 on page 4, the discrepancies in the percentages of both parameters in Table 2 are not understood. The authors should clarify this.
13) In table 2 it is mentioned that "the values indicated are the smallest (1st cell line) and the largest (2nd cell line)." The authors should clarify the meaning of these two values ​​indicated in Table 2. Why are two values ​​shown? Do they correspond to the best and worst results in experiments of the same condition?
14) Table 2 does not show O-TA and T-TA for strength, why?
15) A bibliographic update with a higher number of recent publications is recommended to highlight the novelty of the work.
16) I recommend a detailed review of the English writing.
Comments on the Quality of English LanguageI recommend a detailed review of the English writing.
Author Response
We truly appreciate your proofreading and your comments. To make it easier to read, we’re going to surround your comments and suggestions in black and write our response in italics.
General Comment :
The work is of interest, and shows some interesting results. However, serious deficiencies have been found in the content of the work that prevent the reviewer from becoming aware of the scope of the work carried out by the authors. The main question to clarify is about the calculations made by the authors. The reviewer understands that the FeetMe® system insoles provide a series of parameters, and the authors perform a statistical study of these parameters in a set of experiments performed by a volunteer. In that case, the recommendation would be the rejection of the work because the following reasons:
- The authors have not developed hardware that would fit into a sensor research publication. The authors use a commercial device: FeetMe® system insoles.
- The authors have not developed or investigated software, algorithms or computational models for parameter calculation. The authors use the parameters obtained with the FeetMe® system insoles software.
- The statistical study performed is more suitable for a clinical research journal, rather than a technical research journal.
I would recommend authors submit the work to a journal of clinical interest, rather than a technical research journal.
On the contrary, if the authors have developed and implemented the equations, models and algorithms that allow the calculation of the force and duration parameters of the gait phases, the reviewer would recommend a Major Revision in which the procedures and algorithms used for the calculation of the parameters must be defined in much greater detail. Additionally, greater clarity should be provided on the calculations and parameters used.
Answer :
We understand the reviewer's comment. This issue has been discussed within our team. We regularly publish research on movement disorders. We felt that the analysis of sensors already on the market could help: (i) the clinical community to explore movement disorders, but also (ii) the engineering community to compare the results obtained by their development with reference sensors. Unfortunately, publishers of clinical journals are not very interested in “technical” work. That's why we submitted our study to the journal sensors. Firstly, because this journal regularly publishes work evaluating the use of sensors already developed. Secondly, because we also develop sensors in our own laboratory and, like your team, are probably confronted with the challenge of comparing our results with equivalent solutions or sensors. We attach as much importance to the metrology of the sensors we develop as to the use for which we think they may be useful. In a clinical setting, change detection is essential to the development of a preventive health policy.
Following the modifications proposed by the two reviewers, we have reworded many parts of the paper, which we think will help to underline the interest of our work.
Specific Comments
Q1. Correct references on page 2, line 43: "or to quantify the effect of therapeutic treatments [4–6[4]"
Thank you for pointing out the reference issue on page 2, line 43. We have corrected the reference to ensure clarity and accuracy.
Q2. Reference missing on page 2, line 84: "[conducted a literature review to determine the feasibility of using inertial measurement units to quantify"
Thank you for highlighting the missing reference on page 2, line 84. We have now added the appropriate reference to ensure the information is properly cited and clear.
Q3. Put the reference [8] in the correct place on page 3, line 125: "the sensors in each cell is less than 30 at time (t + 100ms). [8] All the pressure sensors in"
Thank you for pointing out the incorrect placement of the reference [8] on page 3, line 125. We have now corrected the placement to ensure proper citation and clarity.
Q4. The authors mention the following on page 3, lines 141-142: "All the calculations were implemented using custom Python scripts, available online 141 (https://github.com/NathanMartinCOE/semelle_connecte)."
To facilitate understanding of the code and data, I recommend including an explanatory text document in the root directory, such as Readme.txt or Readme.docx
Thank you for your valuable suggestion. We have added a README.md file to the root directory to facilitate understanding of the code.
Q5. Section "2.2. Material: Insole System" describes the processing carried out on the signals. The authors must clarify whether said processing is carried out by the functions of the FeetMe® system insoles, or by functions developed specifically by the authors. If they have been implemented by the authors, the procedures and algorithms must be defined in greater detail.
Thank you for your comment. We would like to clarify that all signal processing mentioned in the text is carried out by the functions of the FeetMe® system insoles, and not by any functions developed by the authors. Here is the revised text for improved clarity:
“All the signal processing described above is performed by the internal algorithms of the FeetMe® system insoles as described by Jacobs et al. [8].”
Q6. The authors indicate on page 3, lines 141-142: "All the calculations were implemented using custom Python scripts". The authors should clarify whether the Python scripts correspond to the calculation of the statistics associated with the forces or durations of the phases in the gait, or instead, they correspond to the calculation of forces and durations of the gait phases, as briefly described in section "2.2. Material: Insole System".
Thank you for your comment. We have revised the paragraph in the manuscript to clarify the data processing workflow:
“All the raw data from the FeetMe® insoles were collected and processed using Py-thon to create a database stored in a CSV file. A custom Python module, containing vari-ous classes and functions for calculating the parameters listed in Table 2, was developed and is available on GitHub (https://github.com/NathanMartinCOE/semelle_connecte). All statistical analyses were performed using R software.”
Q7. In relation to the parameters of section "2.3. Procedure and Data analysis":
- the ground reaction force in the vertical axis,
- the duration of the support phase,
- the duration of the single support phase,
- the duration of the double support pase,
- the duration of the oscillating phase from among all the available parameters
The authors should clarify whether these parameters are estimated by FeetMe® system insoles, or by functions developed by the authors. In the latter case, they must precisely specify the algorithms/models used (see previous comments).
Thank you for your comment. We have revised the paragraph in the manuscript to clarify that:
“All the parameter recordings, provided by the FeetMe® system, are accessible from two comma-separated values files”
Q8. In section "2.3. Procedure and Data analysis" these parameters are mentioned:
- the ground reaction force in the vertical axis,
- the duration of the support phase,
- the duration of the single support phase,
- the duration of the double support phase,
- the duration of the oscillating phase from among all the available parameters
However, Table 2 indicates these parameters:
- Mean force during single stance phase (kg.cm-²)
- Stance duration (ms)
- Single stance duration (ms)
- Double stance duration (ms)
- Swing duration (ms)
Authors should use common terminology to avoid mistakes (They use different terms for the same parameters). In addition, it is advisable to include a short description of the meaning of each parameter.
Thank you for your comment. We have addressed your concerns by adopting a uniform terminology throughout the manuscript to avoid any confusion. Additionally, we have included brief descriptions of each parameter to ensure clarity for the readers:
“We specifically selected:
- Mean force during single stance phase: the ground reaction force in the vertical axis, representing the force exerted by the ground on a body in contact with it,
- Stance duration: the duration of the support phase, representing the time dur-ing which the foot is in contact with the ground supporting the body's weight),
- Single stance duration: the duration of the single support phase, representing the time when only one foot is in contact with the ground while the other foot is in the air,
- Double stance duration: the duration of the double support phase, representing the time when both feet are simultaneously in contact with the ground,
- Swing duration: the duration of the oscillating phase, representing the time when the foot is swinging through the air between ground contacts from among all the available parameters.”
Q9. The number of steps and the duration of each test are mentioned in the work, but the number of tests performed for each type of test is not mentioned. This must be clarified.
Thank you for your observation. To clarify, for each walking condition, three 6-minute walk tests (TDM6) were performed to limit intra-individual variability:
“For each walking condition, three TDM6 were performed in order to limit intra-individual variability.”
Q10. A more detailed and precise description of all calculations must be made. For example: For the calculation of SEM related to "stance duration" perhaps the authors want to say the following: for each gait cycle the stance duration is calculated, and for the entire 6-minute experiment the mean value and variance of the stance duration are calculated. Finally, it is understood that the formula on line 147 on page 4 is used, making a square mean of the standard deviation of all the tests corresponding to a condition. It's right? The procedure must be clarified and specified in detail for all calculated parameters.
11) In relation to the previous question: Table 2 indicates the mean value and standard deviation of the parameters, SEM and MDD95. The authors should clarify precisely how these parameters are calculated.
Thank you for your comment. We appreciate your suggestion for providing a more detailed and precise description of our calculations. Here is a clarified and detailed explanation of our procedure for calculating the Standard Error of Measurement (SEM) and the Minimal Detectable Difference at 95% confidence (MDD95) for our parameters:
Data Collection:
During the 6-minute walk tests (TDM6), all parameters are recorded for each gait cycle.
This results in a series of values for each parameter over the duration of the experiment.
Standard Deviation Calculation:
For each test, the standard deviation of the parameter values is computed.
This step involves calculating the standard deviation for each subject across the different tests.
Root Mean Square Average (RMSA):
The squared standard deviations are averaged across all tests, and then the square root of this average is taken.
This gives us the RMSA, which is used to compute the SEM.
SEM Calculation:
The SEM for each leg (left and right) is calculated by averaging the RMSA values for all tests.
MDD95 Calculation:
The MDD95 is then calculated using the formula:
Q12. Since SEM and MDD95 maintain a relationship according to the equation on line 158 on page 4, the discrepancies in the percentages of both parameters in Table 2 are not understood. The authors should clarify this.
Thank you for your comment. We have reviewed the values in Table 2 and confirmed that there are no discrepancies. The relationship between SEM and MDD95, as described by the formula has been correctly applied. The percentages presented in the table accurately reflect this relationship.
Q13. In table 2 it is mentioned that "the values indicated are the smallest (1st cell line) and the largest (2nd cell line)." The authors should clarify the meaning of these two values ​​indicated in Table 2. Why are two values ​​shown? Do they correspond to the best and worst results in experiments of the same condition?
Thank you for your question. In Table 2, the values shown as the smallest (1st cell line) and largest (2nd cell line) SEM values represent the range of variability observed across different conditions. Specifically, these values correspond to the lowest and highest SEM values calculated for each parameter across the various experiments and conditions. These values are not necessarily the best and worst results from the same condition but rather the extremes observed across all tested conditions and subjects. This range provides insight into the variability and reliability of the measurements within the study. We hope this explanation clarifies the meaning of the two values presented in Table 2.
Q14.Table 2 does not show O-TA and T-TA for strength, why?
Calculating an effect size between two ground reaction force (GRF) curves, which are time-series data, is indeed more complex than using traditional effect size metrics like Cohen's d that rely on single summary statistics such as means. For time-series data, especially when dealing with specific points of interest, a different approach is required. Since we had only one participant, the comparison of the time series did not seem relevant to us, as the results would be too influenced by the individual walking pattern. Therefore, we decided not to calculate the effect size or any other comparison on the GRF. This is different from spatiotemporal parameters, which are much less related to an individual's pattern in healthy subjects.
To clarify this, we have added the following paragraph to our Materials and Methods section:
“Since we had only one participant, the comparison of the time series did not seem relevant to us, as the results would be too influenced by the individual walking pattern. Therefore, we decided not to calculate the effect size or any other comparison on the ground reaction forces curves. This is different from spatiotemporal parameters, which are much less related to an individual's pattern in healthy subjects. The spatiotemporal parameters are more robust and less affected by individual variability, making them more suitable for meaningful analysis and comparison in this context.”
Q15. A bibliographic update with a higher number of recent publications is recommended to highlight the novelty of the work.
Thank you for your feedback. Most of the clinical research articles cited in our manuscript are recent, dating from the last five years. The older references primarily relate to foundational methodologies and statistical analysis techniques. We have also added more recent references to reflect current developments in these areas.
Q16. I recommend a detailed review of the English writing.
Thank you for your feedback. We will conduct a thorough review of the English writing to ensure clarity and accuracy throughout the manuscript.

Round 2
Reviewer 1 Report
Comments and Suggestions for Authors
The authors adequately responded to the questions and comments raised. The study is proposed to be published in its present form.